# Self-reported attitudes, knowledge and skills of using evidence-based medicine in daily health care practice: A national survey among students of medicine and health sciences in Hungary

Mónika Csertő[1], Károly Berényi[2], Tamás Decsi[1], Szimonetta Lohner[1]*

1 Cochrane Hungary, Clinical Centre of the University of Pécs, Medical School, University of Pécs, Pécs, Hungary, 2 Department of Public Health Medicine, Medical School, University of Pécs, Pécs, Hungary

* lohner.szimonetta@pte.hu

## Abstract

In order to map attitudes, knowledge and skills related to evidence-based medicine (EBM) in students of medical and health sciences faculties, we performed an online survey during the spring semester 2019 in all medical and health sciences faculties in Hungary. In total, 1080 students of medicine and 911 students of health sciences completed the online questionnaire. The attitude towards EBM was generally positive; however, only a small minority of students rated their EBM-related skills as advanced. There were large differences in the understanding of different EBM-related terms, with 'sample size' as the term with the highest (65%) and 'intention-to-treat analysis' with the lowest (7%) proportion of medical students being able to properly explain the meaning of the expression. Medical students who already participated in some EBM training rated their skills in searching and evaluating medical literature and their knowledge of EBM-related terms significantly better and had a more positive attitude towards using EBM in the practice than students without previous EBM training. EBM trained medical students were more likely to choose online journals (17.5% compared to 23.9%, p<0.05) and professional guidelines (15.4% compared to 6.1%, p<0.001) instead of printed books (33.6% compared to 52.6, p<0.001) as the main source of healthcare information retrieval and used Pubmed/Medline, Medscape and the Cochrane Library to a significant higher rate than students without any previous EBM training. Healthcare work experience (OR = 1.59; 95% CI = 1.01–2.52), conducting student research (OR = 2.02; 95% CI = 1.45–2.82) and upper year university students (OR = 1.65; 95% CI = 1.37–1.98) were other factors significantly influencing EBM-related knowledge. We conclude that the majority of students of medical and health sciences faculties are keen to acquire EBM-related knowledge and skills during their university studies. Significantly higher EBM-related knowledge and skills among EBM trained students underline the importance of targeted EBM education, while parallel increase of knowledge and skills with increasing number of education years highlight the importance of integrating EBM terminology and concepts also into the thematic of other courses.

**Data Availability Statement:** All relevant data are within the paper and its Supporting Information files.

**Funding:** The preparation of this work was supported by the János Bolyai Research Scholarship of the Hungarian Academy of Sciences (https://mta.hu/) for SL (BO/00498/17/5). TD acknowledges funding from the National Research, Development and Innovation Office (https://nkfih. gov.hu), Grant 120193. Publication fees were covered by the University of Pécs, Medical Faculty. The funders had no role in study design, data collection and analysis, decision to publish, or preparation of the manuscript.

**Competing interests:** The authors have declared that no competing interests exist.

## Introduction

Using evidence-based medicine (EBM) in daily medical and health care practice represents an essential element of developed health care systems. Ideally, in countries with evidence-based practice (EBP), the knowledge generated in clinical trials is timely incorporated into clinical guidelines and serves as a pillar of professional bedside decision making [1]. For successful implementation of EBP, ideas of EBM should become an integral part of the thinking of health care providers at all levels; moreover, besides their adequate knowledge of EBM it is also important that health care professionals possess the willingness and ability to use the acquired knowledge in the daily practice, when making actual decisions about the therapy of patients.

Although EBM is now an accepted part of clinical practice, there are still opposing stand-points: while supporters emphasize facilitated and improved healthcare decisions, which result in a smaller variability in quality of health care provided by different practitioners, the critics take the position that EBM is "cookbook medicine", that it is unable to account for individual patient factors and neglects personal professional experiences [2]. Another potential problem is that health care providers are often not properly trained to implement the evidence [3, 4].

A decade ago the Hungarian EBM working group (developed into Cochrane Hungary), was one of the ten partners who participated in the EU EBM TTT project funded by the European Union. Their goal was to harmonise EBM learning and teaching across the European healthcare sector and to encourage trainers to learn effective teaching methods for tutoring application of EBM in various clinical settings [5]. Cochrane Hungary was founded in 2014 with the aim to provide postgraduate training to healthcare practitioners and to support the understanding of the aims of Cochrane and relevance of EBM among various professionals working at all levels of healthcare.

Currently in Hungary, EBM is taught to students of medicine and health sciences only within the framework of facultative courses. However, during the basic, preclinical and clinical modules of healthcare education there are also several courses which do not have EBM training in their main focus. Nevertheless, these courses incorporate the principles of EBM and teach many EBM-related terms.

However, the attitudes, knowledge and skills of future health care provides, i.e. students of medical and health sciences faculties towards EBM have not been investigated so far, although proper theoretical and practical knowledge about EBM is essential for the realisation of EBP in the near future in Hungary.

The current survey was aimed primarily to evaluate the attitudes, knowledge and skills of students of medicine and health sciences shortly before they finish their studies and start to work as a health care professional. In addition, it aimed to compare data to those obtained in similar students right at the beginning of their university studies. The second aim of this survey was to answer the question to what extent participation in an EBM course during the studies of medicine or health sciences can improve using EBM-related knowledge and skills in the daily health care practice and can change attitudes of students of medicine and health sciences towards evidence based medicine.

## Methods

### Study design and setting

This cross-sectional survey was conducted online between February and May 2019 at every Hungarian medical and health sciences faculties.

### Participants

All medical students studying in one of the four medical faculties in Hungary–namely, 1) University of Pécs, Medical School, Pécs; 2) Semmelweis University, Faculty of Medicine,

Budapest; 3) University of Debrecen, Faculty of Medicine, Debrecen and 4) University of Szeged, Faculty of Medicine, Szeged–were eligible to participate in this survey. All students studying health sciences in seven institutions–namely, 1) University of Pécs, Faculty of Health Sciences, Pécs; 2) Semmelweis University, Faculty of Health Sciences, Budapest; 3) University of Debrecen, Faculty of Health, Nyíregyháza; 4) University of Szeged, Faculty of Health Sciences and Social Studies, Szeged; 5) University of Miskolc, Faculty of Health Care, Miskolc; 6) Gál Ferenc College, Faculty of Health and Social Sciences, Gyula; 7) Széchenyi István University, Petz Lajos Institute of Health and Social Studies, Győr–were also eligible to participate in this survey. Although there are medical and health sciences programs available in English and German at these Hungarian universities, in the frame of the present survey we wanted to obtain information about attitudes, knowledge and skills of Hungarian students studying in the Hungarian programs. Therefore, questionnaires were mailed only to these students, in the Hungarian language. (It is to be noted that due to administrative and financial reasons there are no Hungarian students that participate in the English or German programs of the Hungarian universities). No further exclusions were made.

## Questionnaire and outcomes

The questionnaire was developed by SL using ideas from similar questionnaires [6–11]. The content of these questionnaires was adapted to the target population of this survey and own teaching experience was also incorporated.

The questionnaire was divided into four main parts. The first part included questions regarding the background of the participating student filling in the questionnaire, including the name of the University, the class (year of studies), information on a background with practical work in health care, participation in research activity as member of the Scientific Students' Associations or having a close family member working in health care. The question regarding the participation in a course where EBM was taught was listed among the background questions, therefore students were not aware that this question was one of the main outcomes of the study. Students were also asked which source (printed and online resources, books, journals, professional guidelines etc.) do they consider as their main source of healthcare information retrieval and which search engines have they already used for the retrieval of health care information.

In the second part, students had to self-evaluate their EBM skills, i.e. how experienced they are in identifying patient-relevant questions, locating relevant scientific literature, using online databases for searching and in critical appraisal of already located scientific literature.

In the third part of the questionnaire, important terms of EBM were listed and students had to self-evaluate their knowledge on a 5-point categorical scale. The five ratings were: (1) I understand and I could explain to others; (2) Some understanding; (3) I do not understand, but would like to understand; (4) I do not understand, but I think, it wouldn't be helpful to me to understand; (5) No idea about this.

In the fourth part, attitudes towards using EBM in their future work as a health care professional were evaluated. Statements on the importance of EBM for the practical work and for patients to receive the optimal treatment were listed, and students had to evaluate on a 5-point scale ranging from strongly disagree to strongly agree about their judgements. Statements included also considerations whether evidence-based healthcare incorporates the personal expertise of physicians and the views and preferences of patients, and what extent of burden the application of EBM might mean to health care professionals in the daily routine patient care.

## Pilot testing of the survey

A pre-test was done in a small group (n = 8) of medical students in order to make sure that the study population understood the questions. Study team members and students discussed questions in detail and questions were reformulated, if this was found to be necessary.

## Recruitment, survey administration and data collection

Students were invited to participate in the survey via internal mailing systems of the universities. In the inviting e-mails they received the information that the survey was conducted by Cochrane Hungary with the main aim to receive information about the incorporation of EBM into the Hungarian medical and health sciences education. No further details were provided.

Moreover, small leaflets containing the title and the QR barcode of the questionnaire were distributed among students. In the cities of Pécs and Budapest an information day was held by the study team, where students received not only QR barcodes, but those students without smartphones were also offered the opportunity to fill in the questionnaire on paper instead of the electronic version.

Students were offered to follow a link to the questionnaire website. On the website they were asked to provide informed consent according to the EU General Data Protection Regulation. Only participants providing informed consent were allowed to fill out the questionnaire. Students of medical or health sciences faculties received different links; their questionnaire differed slightly, mainly in the introductory questions.

To encourage honest and transparent responses of the students, anonymity was ensured. Individual data were identified by assigning a unique identification number based on the time point of filling in the questionnaires.

Specific terms to be evaluated were provided in the questionnaire not only in Hungarian, but in parentheses also in English. Terms were listed in alphabetical order.

Data were captured via a Hungarian electronic surface developed for capturing online questionnaires, storing the data obtained from students and enabling a structured export of collected data to Excel and SPSS (http://online-kerdoiv.com/).

## Data analysis

Data were first exported to Excel, in that one line represented answers of one person. Data were analysed using SPSS version 22 (SPSS INC., Chicago, IL, USA); descriptive statistics were calculated for each item. Outcomes for EBM-trained and non-trained students were compared with Mann-Whitney test after rejecting the null hypothesis of Shapiro-Wilk test of normal distribution, in case of quantitative variables. For variables expressed as percentages, Pearson Chi-square test was used. We explored possible associations between certain baseline variables and the attitudes, knowledge and skills by logistic regression models. All results with a significance level of $p < 0.05$ were considered statistically significant.

## Ethical approval

The study was approved by the Scientific and Research Ethics Committee of the Medical Research Council, Budapest, Hungary (60826-1/2018/EKU). Written consent was obtained from the university leaders to conduct the survey.

## Results

### Participant characteristics

A total of 1080 Hungarian students of medicine and 911 Hungarian students of health sciences participated in the survey, which means approximately 17% of Hungarian medical students and 11% of health sciences students currently studying in Hungary. Their baseline characteristics are presented in **Table 1**.

**Medical students.** About one fourth of the medical students filling in the questionnaire have already participated in an EBM teaching course, most of them during the clinical half of medical training (12.7% of the first-year, 13.6% of the second year, 22.2% of the third year, 37.0% of the fourth year, 34.7% of the fifth year and 31.1% of the sixth year respondents).

Only 4.92% of medical students who received training in EBM found the training course inadequate, while all other students were satisfied with its content. Of the 821 medical students who did not participate in an EBM teaching course yet, 94.4% gave the answer that participation in such a course would be helpful for their later practical work as a medical doctor. AS to the place of the course in the curriculum, 10.38% of medical students thought that an EBM course would be effective in the first two years of medical education, 50.93% would like to

**Table 1. Baseline characteristics of students who completed the online survey in the Hungarian faculties of medicine and health sciences.**

| Variable | Medical faculty students (n = 1080) % | Health sciences faculty students (n = 911) % |
|---|---|---|
| **Location of the university** | | |
| • Budapest | 38.24 | 29.09 |
| • Pécs | 31.57 | 39.63 |
| • Szeged | 18.89 | 9.77 |
| • Debrecen | 11.30 | - |
| • Nyíregyháza | - | 5.38 |
| • Miskolc | - | 9.98 |
| • Gyula | - | 5.38 |
| • Győr | - | 0.77 |
| **Class** | | |
| • 1$^{st}$ year | 19.63 | 34.58 |
| • 2$^{nd}$ year | 19.72 | 23.05 |
| • 3$^{rd}$ year | 16.30 | 19.32 |
| • 4$^{th}$ year | 13.52 | 15.04 |
| • 5$^{th}$ year/ MSc 1$^{st}$ year | 16.02 | 5.16 |
| • 6$^{th}$ year/ MSc 2$^{nd}$ year | 14.81 | 2.85 |
| **Gender (male %)** | 37.22 | 11.96 |
| **Practical experience** (worked at least 1 year in health care) | 10.1 | 25.14 |
| **Participating in student research** as member of the Scientific Students' Associations | 34.54 | 9.66 |
| **Near family member** (parent, sibling, spouse) **working in health care** | 36.57 | 39.96 |
| **Frequency of reading professional journals** | | |
| • Daily | 2.87 | 5.60 |
| • Weekly | 21.57 | 25.25 |
| • Monthly or less frequent | 53.61 | 54.88 |
| • Never | 21.94 | 14.27 |
| **Private computer** | 98.80 | 98.24 |
| **Access to internet** | 99.72 | 99.74 |
| **Free internet access** | 65.56 | 57.52 |
| **Participated in a teaching course with EBM training** | 23.98 | 30.08 |

have such a course during the third or fourth year of medical studies, while 38.69% of the respondents answered that they would find a training in EBM effective during the fifth or sixth year of education.

Out of the 1080 medical students, 695 declared that they are reading both Hungarian and English medical sources, while 120 students were reading medical resources not only in English and Hungarian, but also in other foreign languages. While 243 medical students answered that they prefer reading medical literature in Hungarian, only a small minority, i.e. 22 students declared their preference of reading medical literature in English.

**Students of health sciences.** Most of the students of health sciences filling in the survey questionnaire were participating in BSc education, with diverse specialisations. BSc specialisations representing at least 1 percent of participants were as follows: physiotherapists (27.97%), nurses (16.31%), dieticians (10.76%), paramedic officers (9.51%), health visitors (8.15%), medical diagnostic assistants (4.30%), midwifes (4.08%), health tourism managers (3.51%), public health supervisors (3.28%), radiographers (2.38%), recreation and health promotion managers (1.13%). Master specialisations representing at least 1% of participating students of health sciences were: nurses (1.59%), physiotherapists (1.25%), teacher of health sciences and health cares (1.13%) and nutritionists (1.02%).

Out of the 911 students of health sciences filling in the online questionnaire, 274 already participated in an EBM teaching course: 19.4% during the first-year, 26.7% during the second year, 34.7% during the third year and 38.7% during the fourth year among BSc students as well as 58.9% of the MSc students. The large majority of these students, i.e. 94.62% found the course useful for their later work as a health care professional. The vast majority of non-participants (95.06%) would find a training in EBM helpful for their later professional work (36.79% with preference during the 1st or 2nd year of education, while 63.24% with preference during the third or fourth year).

Among participating health sciences faculty students, 46.10% answered reading scientific literature both in the Hungarian and English language, while 45.44% answered reading scientific literature only in the Hungarian language. A small minority (7.35%) of students of health sciences reported reading medical resources also in other foreign languages, while only a very small minority (1.1%) preferred scientific literature in English.

## Questionnaire characteristics

Of the 1991 questionnaires only 7.8% were filled out by the students on paper; the study team members converted these questionnaires into electronicversion.

The consistency testing demonstrated good internal consistency for both the skills questions (Cronbach's alpha = 0.85) and the knowledge-evaluating part (Cronbach's alpha = 0.89), and acceptable internal consistency for the attitude-evaluating part (Cronbach's alpha = 0.71).

## Self-reported skills in EBM

The majority of medical students rated the following skills as average: finding medical literature, searching in online databases, critical appraisal of papers on clinical research and identifying patient-relevant clinical questions. Majority of medical students reported limited experience in critical appraisal of available scientific literature, while ability to identify knowledge gaps were reported to be poor (**Table 2**). Only a minority (under 10% of medical students for all the investigated categories) reported having advanced EBM-related skills (**Table 2**).

Medical students who participated in an EBM course rated all the six items of their skills in searching and evaluating medical literature significantly better than students who did not receive training in EBM (**Table 2**). However, this difference was not as marked in the subgroup

**Table 2. Responses on a 5-point scale to the question: "How would you rate your skills in the following areas?".**

| | Poor | Limited experience | Average | Above average | Advanced | Students with EBM training | Students without EBM training | |
|---|---|---|---|---|---|---|---|---|
| | (1) | (2) | (3) | (4) | (5) | Mean score (SD) | Mean score (SD) | p |
| **Medical students** (n = 1080) | | | | | | **n = 259** | **n = 821** | |
| **Locating professional literature** | 5.37% | 18.06% | 44.26% | 25.83% | 6.48% | 3.46 (0.89) | 2.99 (0.94) | <0.001 |
| **Searching online databases** | 5.19% | 16.94% | 40.09% | 28.89% | 8.89% | 3.42 (1.01) | 3.12 (0.98) | <0.001 |
| **Critical appraisal of a scientific publication reporting findings from clinical research** | 22.96% | 31.2% | 30.09% | 13.33% | 2.41% | 2.79 (1.02) | 2.29 (1.03) | <0.001 |
| **Identifying knowledge gaps in practice (fields where not enough scientific literature is available to answer a specific clinical question)** | 38.89% | 32.31% | 20.46% | 6.76% | 1.57% | 2.34 (1.08) | 1.89 (0.95) | <0.001 |
| **Critical appraisal of available scientific literature** | 16.2% | 30.37% | 30.56% | 19.35% | 3.52% | 2.94 (1.09) | 2.54 (1.05) | <0.001 |
| **Identifying patient-relevant clinical questions** | 6.02% | 16.11% | 36.39% | 32.5% | 8.98% | 3.45 (1.01) | 3.15 (1.01) | <0.001 |
| **Health sciences faculty students** (n = 911) | | | | | | **n = 274** | **n = 637** | |
| **Locating professional literature** | 4.28% | 14.49% | 50.05% | 24.81% | 6.37% | 3.23 (0.83) | 3.09 (0.92) | 0.06 |
| **Searching online databases** | 3.40% | 11.42% | 42.15% | 29.97% | 13.06% | 3.42 (0.89) | 3.36 (1.00) | 0.47 |
| **Critical appraisal of a scientific publication reporting findings from clinical research** | 20.97% | 35.13% | 33.48% | 8.45% | 1.98% | 2.55 (0.97) | 2.22 (0.94) | <0.001 |
| **Identifying knowledge gaps in practice (fields where not enough scientific literature is available to answer a specific clinical question)** | 36.33% | 34.80% | 22.83% | 4.50% | 1.54% | 2.20 (0.95) | 1.87 (0.93) | <0.001 |
| **Critical appraisal of available scientific literature** | 13.94% | 30.63% | 37.32% | 15.48% | 2.63% | 2.84 (0.94) | 2.50 (0.99) | <0.001 |
| **Identifying patient-relevant clinical questions** | 3.62% | 14.82% | 38.31% | 33.59% | 9.66% | 3.58 (0.96) | 3.17 (0.95) | <0.001 |

of first and second year medical students, where such students participating in an EBM course rated only their skills in critical appraisal of the content of a scientific publications (p = 0.007) and in identifying knowledge gaps (p = 0.025) significantly better as compared to student who have not yet participated in such a course. Among third and fourth year medical students, all the six investigated skills (**Table 2**) were evaluated as significantly better in the subgroup of EBM course participants as compared to non-participants. The same was also true for the subgroup of fifth and sixth year medical students.

The distribution of answers to certain questions was similar among students of health sciences to that seen in the case of medical students, with the majority of students of health sciences rating their skills as average in locating professional literature, searching in online databases, in critical appraisal of papers on clinical research and in identifying patient-relevant clinical questions. Students had limited experience in critical appraisal of available scientific literature and rated their skills in identifying knowledge gaps as poor (**Table 2**). With the exception of searching in online databases, students of health sciences with EBM training had significantly higher ratings than had EBM non-trained students.

## Sources and methods of healthcare information retrieval

The percentage distribution of answers to the question "Which source would you rate as the primary source of healthcare information retrieval?" among medical students who either participated or not in an EBM course are compared on **Fig 1**. Medical students not yet trained in EBM were significantly more likely to choose printed books as the main source of healthcare information retrieval, while medical students who already participated in an EBM course choose online journals and professional guidelines to a significantly higher extent compared to those who did not participate in EBM course.

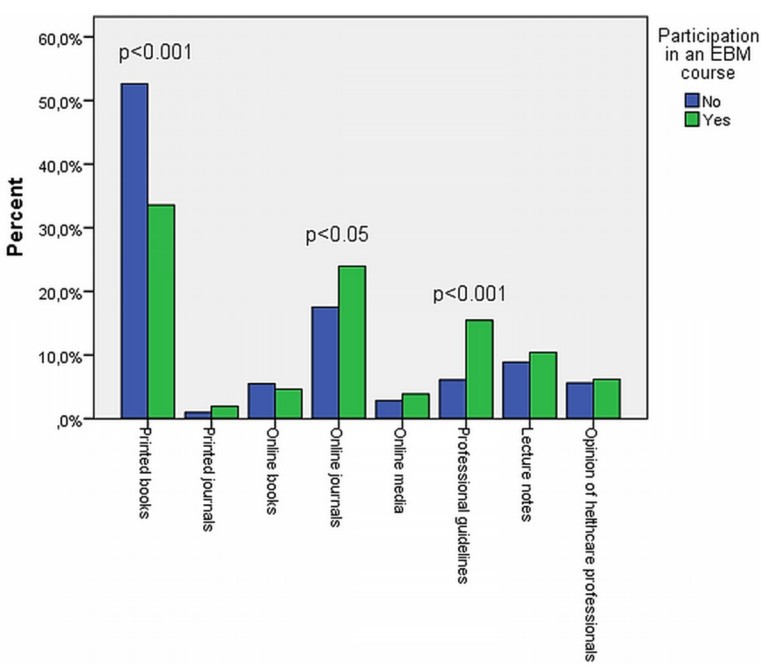

**Fig 1. The opinion of Hungarian medical students regarding the most important source of healthcare information retrieval.**

Among students of the faculty of health sciences who were trained in EBM choose printed books to a significantly lower extent (28.8% vs 37.8%; p = 0.01) and professional guidelines to significantly higher extent (10.2% vs 3.6%; p<0.001) than those students who did not participate in EBM course. No other significant difference was seen between the two groups.

Search engines used for healthcare information retrieval are shown in **Table 3**. (Here results were available for 1465 students only, as due to a technical problem not all students answered the question "Which search engines have you already used for healthcare information retrieval?" as a multiple choice, but as a single choice question. Only data from students answering the question as multiple choice were analysed.)

**Table 3. Popularity of different search engines among Hungarian students of medicine and health sciences faculties.**

| | |
|---|---|
| **Medical students** (n = 697) | |
| Google | 94.12% |
| Google scholar | 27.12% |
| Wikipedia | 72.45% |
| Pubmed/ Medline | 63.99% |
| Medscape | 22.38% |
| Cochrane Library | 5.02% |
| **Health sciences faculty students** (n = 768) | |
| Google | 90.63% |
| Google scholar | 20.57% |
| Wikipedia | 52.60% |
| Pubmed/ Medline | 49.35% |
| Medscape | 17.45% |
| Cochrane Library | 4.69% |

Google was the most popular search engine, followed by Wikipedia and Pubmed/Medline. Medscape and especially the Cochrane Library were used by only a minority of students for the retrieval of scientific literature (**Table 3**).

When conducting a subgroup analysis comparing EBM trained and non-trained medical students, we found that Pubmed/Medline (84.0% vs. 57.9%, [EBM trained vs. EBM non-trained], p<0.001), Medscape (36.2.0% vs. 18.2%, p<0.001), and the Cochrane Library (12.3% vs. 2.8%, p<0.001) were used to a significantly higher extent for healthcare information retrieval by EBM trained than non-trained medical students. There were no significant differences in the use of Google (92.6% vs. 94.6%, p = 0.35), Google Scholar (30.7% vs. 26.0%, p = 0.27) and Wikipedia (73.8% vs. 68.1%, p = 0.16) between the EBM trained and non-trained medical students.

Although the use of Pubmed, Medscape and Cochrane Library were generally lower among students of health sciences than among medical faculty students, subgroup analysis comparing EBM trained and non-trained health sciences students had comparable results to those seen in the case of medical students: Pubmed/Medline (62.7% vs. 43.6%, [EBM trained vs. EBM non-trained], p<0.001), Medscape (25.3.0% vs. 14.0%, p<0.001), and Google scholar (26.6% vs. 17.9%, p = 0.01) were used to a significantly higher extent by EBM trained than non-trained health sciences faculty students, while there were no significant differences in the use of Google (90.6% vs. 90.7%, p = 1.00) and Wikipedia (51.9% vs. 52.9%, p = 0.81). Among students of the faculty of health sciences, the use of the Cochrane Library did not significantly differ between EBM trained and non-trained students (6.4% vs. 3.9%, p = 0.14).

## Knowledge of EBM-related terms

There were large differences in the understanding of different EBM-related terms (**Table 4**). The most known term among medical students was 'sample size'; about two third of medical students answered that they could explain meaning of the term to others. In contrast, only 7% of medical students thought that they could explain the meaning of the term 'intention-to-treat analysis'. When evaluating answers as if they were scores on a 5-point-scale and comparing the range of scores between EBM-trained and non-trained medical students, we found that those participating in an EBM course rated their knowledge regarding EBM-related terms to be significantly better than students who did not receive a training in EBM yet (with p<0.001 for all investigated terms).

We also calculated a mean score based on the 13 scores (listed in **Table 4**) evaluating knowledge of individual students and conducted a multifactorial logistic regression analysis to reveal factors having an influence on better or worse scores (i.e. a mean score higher or lower than 2.0). Healthcare work experience (OR = 1.59; 95% CI = 1.01–2.52, p = 0.048), conducting student research as member of the Scientific Students' Associations (OR = 2.02; 95% CI = 1.45–2.82, p<0.001), upper year university students (OR = 1.65; 95% CI = 1.37–1.98, p<0.001) and participation in an EBM teaching course (OR = 3.32; 95% CI = 2.32–4.76, p<0.001) proved to have a significant positive influence on the knowledge of EBM related terms among medical students, while gender (OR = 1.22; 95% CI = 0.93–1.61, p = 0.15) and having a close family member working in healthcare (OR = 0.96; 95% CI = 0.73–1.26, p = 0.75) had no significant effect.

Among students of health sciences, the most known term was 'case study', while the least known was 'confidence interval' (**Table 4**). Also health sciences faculty students with EBM training rated their knowledge on most of the EBM-related terms significantly better than students without EBM training, although there was no significant difference in the knowledge of trained and non-trained students in case of the terms 'lost to follow-up' (p = 0.15) and 'number

**Table 4. Self-reported understanding of evidence-based healthcare-related terms among Hungarian medical and health sciences faculty students.**

| | I understand and I could explain to others | Some understanding | Do not understand, but would like to understand | Do not understand, but I think, it wouldn't be helpful to me to understand | No idea about this |
|---|---|---|---|---|---|
| **Medical students** (n = 1080) | | | | | |
| Evidence-based medicine | 43.33% | 38.98% | 15.83% | 0.28% | 1.57% |
| Intention-to-treat analysis | 6.96% | 25.65% | 60.65% | 3.43% | 3.33% |
| Sample size | 65.09% | 25.19% | 6.94% | 0.83% | 1.94% |
| Case study | 59.07% | 33.61% | 4.91% | 0.83% | 1.57% |
| Cohort study | 30.74% | 30.74% | 33.70% | 2.04% | 2.78% |
| Confidence interval | 33.61% | 37.22% | 23.80% | 3.80% | 1.57% |
| Controlled clinical study | 44.44% | 37.04% | 16.48% | 0.74% | 1.30% |
| Lost to follow-up | 37.59% | 31.67% | 26.20% | 2.04% | 2.50% |
| Meta-analysis | 25.46% | 24.44% | 43.70% | 3.70% | 2.69% |
| NNT (number needed to treat) | 14.26% | 27.13% | 51.94% | 3.43% | 3.24% |
| Randomisation | 53.61% | 31.57% | 12.50% | 1.02% | 1.30% |
| Practical guideline | 58.15% | 30.74% | 9.35% | 0.56% | 1.20% |
| Systematic review | 28.80% | 38.70% | 28.89% | 1.48% | 2.13% |
| **Health sciences faculty students** (n = 911) | | | | | |
| Evidence-based medicine | 39.85% | 39.96% | 16.47% | 1.43% | 2.31% |
| Intention-to-treat analysis | 10.10% | 33.04% | 48.85% | 4.28% | 3.73% |
| Sample size | 53.35% | 29.09% | 11.96% | 2.31% | 3.29% |
| Case study | 54.77% | 31.17% | 9.11% | 1.65% | 3.29% |
| Cohort study | 16.90% | 27.11% | 47.31% | 3.29% | 5.38% |
| Confidence interval | 8.89% | 21.62% | 57.08% | 6.59% | 5.82% |
| Controlled clinical study | 31.94% | 38.97% | 23.82% | 2.31% | 2.96% |
| Lost to follow-up | 44.24% | 28.76% | 22.50% | 1.98% | 2.52% |
| Meta-analysis | 18.66% | 24.70% | 48.74% | 3.62% | 4.28% |
| NNT (number needed to treat) | 9.44% | 21.41% | 58.84% | 4.61% | 5.71% |
| Randomisation | 41.93% | 32.27% | 21.41% | 1.76% | 2.63% |
| Practical guideline | 54.34% | 33.26% | 9.22% | 1.32% | 1.87% |
| Systematic review | 33.59% | 39.30% | 21.30% | 2.85% | 2.96% |

needed to treat' (p = 0.05). Upper year university students (OR = 1.47; 95% CI = 1.05–2.05, p<0.05) and participation in an EBM teaching course (OR = 1.69; 95% CI = 1.47–1.93, p<0.001) were factors that significantly influenced EBM-related knowledge.

## Attitudes towards using EBM in health care practice

All medical students agreed that EBM is important for the practical work of physicians and wished to improve their skills in applying EBM (**Table 5**). However, students trained in EBM were more likely to answer "strongly agree" instead of "agree", which resulted in significant difference between the EBM-trained and non-trained groups in the 8 out of 11 parameters compared (**Table 5**). The degrees of agreement with the statements that "EBM is important for patients to receive the optimal treatment" and that "EBM facilitates decisions about individual

**Table 5. Response frequency and means of ratings to the question: "On a scale ranging from 'strongly disagree' to 'strongly agree' how would you rate your opinion about the following statements?"** among Hungarian medical students (n = 1080).

| | Strongly disagree | Disagree | Neutral | Agree | Strongly agree | Students with EBM training (n = 259) | Students without EBM training (n = 821) | |
|---|---|---|---|---|---|---|---|---|
| | (1) | (2) | (3) | (4) | (5) | mean range (SD) | mean range (SD) | p |
| Evidence based medicine (EBM) is important for the practical work of physicians | 0.56% | 0.83% | 12.87% | 45.00% | 40.74% | 4.50 (0.68) | 4.17 (0.75) | <0.001 |
| During my studies, I would like to improve my skills in applying EBM during my practical work as a medical professional | 0.28% | 0.83% | 16.76% | 49.35% | 32.78% | 4.31 (0.76) | 4.08 (0.72) | <0.001 |
| EBM is important for patients to receive the optimal treatment | 0.28% | 0.74% | 14.54% | 49.91% | 34.54% | 4.42 (0.67) | 4.10 (0.71) | <0.001 |
| EBM facilitates decisions about individual patient's care | 0.37% | 0.74% | 14.35% | 48.06 | 36.48% | 4.46 (0.69) | 4.11 (0.72) | <0.001 |
| EBM considers the personal expertise of physicians | 2.78% | 18.43% | 43.80% | 26.39% | 8.61% | 3.18 (1.06) | 3.20 (0.89) | 0.83 |
| EBM considers views and preferences of patients regarding their own therapy | 4.07% | 23.43% | 44.07% | 22.04% | 6.39% | 2.94 (1.08) | 3.06 (0.88) | 0.07 |
| It is important to incorporate research results into healthcare practice | 0.28% | 0.19% | 5.65% | 42.87% | 51.02% | 4.53 (0.63) | 4.42 (0.64) | 0.01 |
| All types of studies are of equal value | 19.07% | 53.61% | 19.26% | 7.13% | 0.93% | 2.00 (0.90) | 2.23 (0.83) | <0.001 |
| EBM means an unrealistic burden to health care professionals in the daily routine patient care | 7.96% | 42.13% | 42.13% | 6.94% | 0.83% | 2.41 (0.88) | 2.53 (0.74) | 0.01 |
| Textbooks are the most optimal source of information, when a question regarding the care of patients should be answered | 3.7% | 29.26% | 38.7% | 25.74% | 2.59% | 2.84 (0.93) | 2.98 (0.88) | 0.06 |
| As a future healthcare practitioner, I find life-long learning as vital | 0.37% | 0.93% | 5% | 30.93% | 62.78% | 4.66 (0.56) | 4.51 (0.70) | 0.01 |

patient's care" were also significantly higher in the EBM-trained medical student group than in those without EBM training.

The majority of medical students were unable to decide whether EBM considers also the personal expertise of physicians as well as the views and preferences of patients regarding their own therapy; in this context there were no differences between EBM-trained and non-trained medical students. Medical students were also unsure whether textbooks are the most optimal source of information when questions regarding the care of individual patients should be answered (Table 5). Life-long learning was seen as very important in both groups, but scores representing the strength of agreement were significantly higher among EBM trained medical students (Table 5).

Among students of health sciences there were 6 out of 11 statements with significant difference between the opinion of EBM-trained and non-trained students (Table 5).

## Discussion

### Summary of findings

The present study attempted to provide an overall picture about the extent to which concepts of EBM are incorporated into medical and health sciences education in Hungary and to answer the question whether evidence-based education in the present form is effective enough to improve skills and knowledge and build a generally positive attitude towards EBM among students.

In the international scientific literature several studies are available that assess the attitudes, knowledge and skills of medical and healthcare professionals [7, 12–18] and describe generally

positive attitudes, however with skills of very different level in using EBM in practice. Moreover, in countries, where EBM courses are already incorporated into the curriculum, surveys have assessed the attitudes, knowledge and skills of both educators [6] and educated students [19–22]. These studies consistently reported positive attitudes toward EBM among undergraduate students participating in EBM education.

The uniqueness of our study is that it compares attitudes, knowledge and skills of students who participated in an EBM training course and those who are not EBM course attendees studying in the same institutions and faculties. Therefore we think, that our findings may also be adaptable and useful to countries where, as in Hungary, only a part of students receive focused EBM education or where EBM concepts are just in progress to be introduced into curriculum.

In the present study, the attitude towards EBM was generally positive among both medical and health sciences students; however, only a small minority of students rated their EBM-related skills as advanced, and there were large differences in the understanding of different EBM-related terms. General terms, like 'evidence-based medicine' or 'sample size' were better understood than more specific terms, like 'intention-to-treat analysis', 'confidence interval' or 'number needed to treat'. This difference might be related simply to the fact that general terms are more likely to appear also during the education of subjects other than EBM. These results suggests that a list of EBM-related terms which is constructed at faculty level, handed out and recommended for medical and health sciences faculty students during their studies might improve the transmission of EBM-related knowledge.

Importance of targeted EBM training is strongly underpinned in our study by the results on sources and methods used by students for scientific information retrieval. Of course, for a large majority of university students printed books represent the most important source of information. However, EBM training seems to be an important tool to educate students of the importance of using up-to-date scientific information (e.g. those published in online journals) for supporting healthcare decisions. The significant difference in the use of PubMed/Medline and the more than fourfold difference in the use of the Cochrane Library seen in this study in medical students who participated in EBM courses as compared to those who did not highlight the importance of teaching how to use these data retrieval systems for medical decision making.

Among both medical and health sciences faculty students, upper year students and participating in research activities were important factors contributing to EBM-related knowledge. Because the percent of EBM-trained student was also increasing in parallel with the number of years of studies, this observation might underpin further the important role of incorporating EBM education into other courses besides targeted EBM training.

Students of health sciences were generally characterised by more self-confidence, especially when self-evaluating their EBM-related skills. However, differences between EBM-trained and EBM non-trained students from the faculty of health sciences were less pronounced than in the case of medical students. These slight differences between students of medicine and health sciences might not necessarily reflect the lower effectiveness of EBM training among students of health scientific, but might rather indicate that the number of non-EBM-trained students overestimating their knowledge was higher in our sample among students of the faculties of health sciences.

It is interesting that the large majority of medical students were neutral regarding the statements 'EBM considers the personal expertise of physicians' and 'EBM considers views and preferences of patients' with no significant differences between EBM-trained and non-trained students. This observation indicates that medical students, at least in Hungary, are unsure about the way how EBM should be implemented in the daily practice. There is considerable

potential for improvements to move away from "cookbook medicine" towards a science-based, but individualised medicine that involve both professional expertise and individual patient factors.

## Study limitations

Although attempts were made to maximize the rate of filling out the questionnaire by a representative number of students by sending out invitations and reminders to participate several times, the participation rate from different universities does not fit the proportion of the students studying there. Consequently, students with more active attitudes towards scientific or public life might be overrepresented in the sample. Moreover, first and second year students were more eager to participate in the survey, therefore their opinion might be overrepresented against the opinion of students from upper university years. We cannot fully exclude response bias and should be cautious with self-reported information [23].

In the present survey we have not asked detailed information about the characteristics (e.g. hours, content) of the EBM course attended by the students. Additionally, the possible differences in EBM education among faculties and specialisations were not studied and their impact were not analysed.

During the university studies there is a unique opportunity to form attitudes of future healthcare providers and to pass over EBM-related knowledge, however in our findings, i.e. attitudes, knowledge and skills of undergraduate medical and health sciences faculty students do not necessarily reflect the real use these students will make of EBM later as a health professional.

## Conclusions

Substantial proportion of students of the medical and health sciences faculties would like to acquire EBM-related knowledge and skills during their university studies. Although the attitude towards EBM is generally positive, only a small minority of students rated their EBM-related skills as advanced in the present survey. Self-reported EBM-related knowledge and skills are higher among students who already received an EBM-training, an observation which underlines the role of targeted EBM education in both medical and health sciences education. Targeted EBM training seems to be more effective following the second education year, when medical students have already acquired basic knowledge in medicine. Increased EBM-related knowledge and skills among higher year medical students highlight the importance of integrating EBM concepts also into other courses of the basic, preclinical and clinical modules.

## Supporting information

**S1 Questionnaire.**
(XLS)

**S2 Questionnaire.**
(XLS)

**S3 Questionnaire.**
(PDF)

**S4 Questionnaire.**
(PDF)

## Acknowledgments

We would like to thank Kinga Amália Sándor-Bajusz for reviewing our manuscript for language accuracy.

## Author Contributions

**Conceptualization:** Szimonetta Lohner.

**Data curation:** Mónika Csertő.

**Formal analysis:** Károly Berényi.

**Investigation:** Mónika Csertő.

**Methodology:** Szimonetta Lohner.

**Project administration:** Mónika Csertő.

**Supervision:** Szimonetta Lohner.

**Visualization:** Tamás Decsi, Szimonetta Lohner.

**Writing – original draft:** Szimonetta Lohner.

**Writing – review & editing:** Mónika Csertő, Károly Berényi, Tamás Decsi, Szimonetta Lohner.

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
