## [Decision Letter · Decision Letter 0]

14 Oct 2019

PONE-D-19-21546

Self-reported attitudes, knowledge and skills of using evidence-based medicine in daily health care practice: a national survey among students of medical and health sciences faculties in Hungary

PLOS ONE

Dear Dr Lohner,

Thank you for submitting your manuscript to PLOS ONE. After careful consideration, we feel that it has merit but does not fully meet PLOS ONE’s publication criteria as it currently stands. Therefore, we invite you to submit a revised version of the manuscript that addresses the points raised during the review process.

Please address all issues raised by both reviewers in a response letter and in the revised version (if you find necessary make changes as suggested). In particular, the issue of bias in this type of survey,  make the data publicly available, make it shorter by eliminating  text redundancy  and re-write the conclusion. Whether a written request from all universities is necessary.

We would appreciate receiving your revised manuscript by Nov 28 2019 11:59PM. To enhance the reproducibility of your results, we recommend that if applicable you deposit your laboratory protocols in protocols.io, where a protocol can be assigned its own identifier (DOI) such that it can be cited independently in the future. For instructions see: http://journals.plos.org/plosone/s/submission-guidelines#loc-laboratory-protocols

We look forward to receiving your revised manuscript.

Kind regards,

Cesario Bianchi

Academic Editor

PLOS ONE

Journal Requirements:

Additional Editor Comments (if provided):

Dear Dr. Lohner:

Your manuscript was revised by 2 experts that found it of interest. However, there are some issues that need your attention before I could make a decision.

Reviewer 1 is concerned about response bias (self-response) and if the data can be generalized (beyond Hungary). The questionnaire should be presented in a supplement. The reviewer is concerned about the necessity for written permission from all universities involved.

Reviewer 2 would like more detail methodology and found that most parts of the manuscript to be redundant making it too long and difficult to read. The discussion needs to be revised and some references may be added.

There are limitations of the survey, in particular, that was applied only to undergraduate students.

Where is the origin of the questionnaire used (was it validated?) and the data needs to be publicly available.

Reviewers' comments:

Reviewer's Responses to Questions

**Comments to the Author**

1. Is the manuscript technically sound, and do the data support the conclusions?

Reviewer #1: Yes

Reviewer #2: Yes

2. Has the statistical analysis been performed appropriately and rigorously? 

Reviewer #1: Yes

Reviewer #2: Yes

3. Have the authors made all data underlying the findings in their manuscript fully available?

Reviewer #1: Yes

Reviewer #2: No

4. Is the manuscript presented in an intelligible fashion and written in standard English?

Reviewer #1: Yes

Reviewer #2: Yes

5. Review Comments to the Author

Reviewer #1: Please see attached document for comments for the manuscript. Thank you for the opportunity to review.

Reviewer #2: Overall, this is a well conducted and adequately reported survey. Some additional details would be needed for a complete reporting of the methods employed (see below). The text is too long in most sections of the manuscript, with some ideas and phrases repeated several times and I think the paper would be much more readable if it could be shortened.

The main limitation of the study, in my view, is that it included only undergraduate students and, while it is always interesting to know more about knowledge and attitudes towards EBM in health professions, that does not allow us to know what will be the real use those students will make of EBM later, in their daily practice. I think this limitation should be acknowledged in the discussion.

INTRODUCTION

The first part of the introduction states a very idealistic view of EBM in practice. It could be discussed if there is actually any country where “the knowledge generated in clinical trials is timely incorporated into clinical guidelines and serves … [for] decisions made at the bedside”. Most surveys conducted in practicing doctors, nurses and other professions have found that only a minority show a good knowledge of EBM or actually implements EBM in their practice. See references 6 to 11, as well as Oliveri RS, J Eval Clin Pract 2004 ; Heselmans A, BMC Fam Pract 2009 or Ulvenes LV, PLoS One 2009.

METHODS

- Give please, some details about the EBM teaching students followed: how many hours? only theoretical content? Or it included practical cases?

- Did you derived your questionnaire from any questionnaire previously published? If so, stated which one. If not, state you entirely developed it and explain how you did it: who created it, how was it developed.

- What you call “Validation of the survey” is not a proper validation of the questionnaire (testing their reliability, accuracy or the construct consistency). It is just a (small) pilot testing of the questionnaire. It should be named such as.

- Some ideas (ex. questionnaire sent by e-mail, English vs Hungarian language) are repeated several times and that makes the text too long.

RESULTS

- Full data of the survey is not made available in any public repository.

- The text is too long. Try to synthetize and shorten it. I’d suggest following a clear structure along all subheadings: give first the results for the overall population, then the (interesting, important) differences between EBM-trained and not trained students. What did EBM-trained students declare more frequently that those not trained, in a nutshell? Do not repeat figures already given in tables, unless they are especially important.

DISCUSSION

- The first two paragraphs have, in my opinion, little added value, as the ideas here have been already said in previous sections. I’d rather begin the summary of findings in the third paragraph. Conversely, the discussion lack a section comparing the results obtained with previuosly published studies.

- Conclusions: I think it would be important to also underline in your conclusions that: 1) A vast majority of students show a very positive attitude towards EBM; 2) However, only a minority of students considered their EBM skills as good or advanced.

6. PLOS authors have the option to publish the peer review history of their article (what does this mean?). If published, this will include your full peer review and any attached files.

Reviewer #1: No

Reviewer #2: Yes: Carmelo Lafuente-Lafuente

---

## [Author Response · Author response to Decision Letter 0]

29 Oct 2019

Response to Academic Editor:

We would like to thank the Editor for all his editorial efforts. 

We addressed all the concerns reviewers have raised. We added additional details to the Methods part of the study report and deleted redundant parts. We added additional references where requested and edited the Discussion section based on Reviewer comments. 

We added the questionnaire as a supplement (both the original questionnaire in Hungarian and an English translation). We have all necessary permissions from the universities, which were required to conduct the survey and to publish data.

Answers to Reviewer 1:

We thank the reviewer for the positive feedback and all the comments and suggestions that helped us considerably to improve the quality of our paper. We really hope that we were able to address the comments and questions adequately. We respond to the specific points raised as follows:

1. The methods are generally well reported, however there are a few omissions which would be useful to clarify: What language was the survey conducted in?; Did any exclusion criteria apply (i.e. students with formal research backgrounds)?; It would be useful to provide a copy of the questionnaire, or at least a full outline of the questions in supplementary format. 

The survey was conducted in Hungarian and no students were excluded (for the language of the survey, please see LL156-158. We now added the information that no exclusions were made, see L104-105).

We now added the survey as supplement, both in original language (Hungarian) and an English language translation.

2. National completion of the survey is a strength since it helps to increase generalisability. To what extent might these results be useful beyond Hungary? - please consider commenting on this in the discussion. 

We included the following sentence into Discussion: “We think, that our findings may also be adaptable and useful to countries where, as in Hungary, only a part of students receive focused EBM education or where EBM concepts are just planned to be introduced into curriculum.”

3. I agree with the study limitations, but these need further elaboration. The key limitation seems to be the role of response bias. I think a fuller discussion about this is required i.e. to what extent is a sample of 17% generalisable to the Hungarian healthcares student population; what is the likely impact of self-responses to survey questions (bias), etc. 

To avoid responding bias, we conducted a small pilot testing to ensure that the formulation of questions was precise and simple enough to be understood by participating students. We tried to formulate each of the question short and clear. The survey was structured in sections and questions had to be answered in a given order by each participant. We tried to cover all potential answers by including options such as “Neutral” and “No idea about this”. We tried to avoid leading questions and – as described in the manuscript – “the question regarding the participation in a course where EBM was taught was listed among the background questions, therefore students did not know that this question was one of the main outcomes of the study”.

As we discussed under Study limitations, students “with more active attitudes towards scientific or public life might be overrepresented in the sample. Moreover, first and second year students were more eager to participate in the survey, therefore their opinion might be overrepresented against the opinion of students from higher study years”. Therefore, it is difficult to estimate the extent to which our results can be generalized, and for this reason, we state that “We cannot fully exclude response bias and should be cautious with self-reported information”.

4. There is clearly a large appetite for students learning the principles of EBM. This study identifies some key attitudes and perceptions, but how might these results be taken forward? I would suggest that this is briefly considered in the Discussion. 

We really hope, that university leaders will understand our conclusion that “Significantly higher EBM-related knowledge and skills among EBM trained students underline the importance of targeted EBM education, while parallel increase of knowledge and skills with increasing number of education years highlight the importance of integrating EBM terminology and concepts also into the thematic of other courses“. As evidence-based medicine is currently taught to Hungarian medical and health sciences faculty students within the frame of facultative courses, a large step forward would be, if EBM training would become reclassified to a compulsory subject.

5. A discretionary comment: Often institutions do not look favourably on centre-specific comparisons (i.e. comparisons of data between Universities) unless permission is sought. This may be different in Hungary. To avoid problems later, I would suggest that the authors check to ensure this is permitted. 

We completely agree. This is why we obtained written consent from all the university leaders in advance to conduct the survey (now we added this information to the manuscript). They were informed also about the fact that scientific results will be published. Nonetheless, we avoided direct comparisons between Universities in the manuscript (the only university-specific information provided is the number of participants).

6. The manuscript is generally written well, the tables are well presented, and the references seem appropriate.

Thank you very much for the supporting comments.

Answers to Reviewer 2:

We would like to thank the Reviewer for all the comments and suggestions, which helped us to improve the quality and transparency of study reporting. We answer the comments and questions raised as follows:

1. The main limitation of the study, in my view, is that it included only undergraduate students and, while it is always interesting to know more about knowledge and attitudes towards EBM in health professions, that does not allow us to know what will be the real use those students will make of EBM later, in their daily practice. I think this limitation should be acknowledged in the discussion.

We think that during the university studies there is a unique opportunity to form attitudes of future healthcare providers and to pass over EBM-related knowledge. The aim of this study was therefore to focus on this specific group and to measure where EBM university education currently stands in Hungary, how effective it is in this area. 

However, we agree with the Reviewer that this does not necessarily reflect the real use students will make of EBM later, so, we added a sentence to the Study limitations part of Discussion to elucidate this.

Our main aim in the frame of the present study was to focus on undergraduate students. However, we plan to make a further survey among medical doctors and healthcare practitioners, to measure also their attitudes, knowledge and skills.

2. The first part of the introduction states a very idealistic view of EBM in practice. It could be discussed if there is actually any country where “the knowledge generated in clinical trials is timely incorporated into clinical guidelines and serves … [for] decisions made at the bedside”. Most surveys conducted in practicing doctors, nurses and other professions have found that only a minority show a good knowledge of EBM or actually implements EBM in their practice. See references 6 to 11, as well as Oliveri RS, J Eval Clin Pract 2004 ; Heselmans A, BMC Fam Pract 2009 or Ulvenes LV, PLoS One 2009.

We added the phrase “Ideally…” to the sentence in question, to express that we aimed to describe the ideal situation of EBP.

3. Give please, some details about the EBM teaching students followed: how many hours? only theoretical content? Or it included practical cases?

Unfortunately, we have not assessed this in detail in the survey. However, currently only facultative EBM courses are available in Hungary, which mainly focus on theoretical aspects.

As this may be considered as a limitation of our survey, we now have added a sentence to the Study limitations: “In the present survey we haven’t asked detailed information about the characteristics (e.g. hours, content) of the EBM course students attended and possible differences in EBM education among faculties and specialisations and their impact were not analysed.”

4. Did you derived your questionnaire from any questionnaire previously published? If so, stated which one. If not, state you entirely developed it and explain how you did it: who created it, how was it developed.

We added the following explanation to the manuscript: “The questionnaire was developed by SL using ideas from similar questionnaires (Young 2016, Bussiers 2015, Terhorst 2016, Parve 2016, Nieminen 2017, Mahmic-Kaknjo 2015), adapting their content to the target population of this survey, incorporating also own teaching experience.” We added all those references to our report, which we used for developing the questionnaire.

5. What you call “Validation of the survey” is not a proper validation of the questionnaire (testing their reliability, accuracy or the construct consistency). It is just a (small) pilot testing of the questionnaire. It should be named such as.

We corrected this misleading expression to “Pilot testing of the survey”, as suggested by the Reviewer.

6. Some ideas (ex. questionnaire sent by e-mail, English vs Hungarian language) are repeated several times and that makes the text too long.

We edited the text and deleted redundant information, as advised by the Reviewer.

7. RESULTS. Full data of the survey is not made available in any public repository.

We have chosen the option to submit full survey databases as Supporting Information Files, which – as we know – will be uploaded to figshare upon publication, so the data will be available in a cross-disciplinary repository.

8. The text is too long. Try to synthetize and shorten it. I’d suggest following a clear structure along all subheadings: give first the results for the overall population, then the (interesting, important) differences between EBM-trained and not trained students. What did EBM-trained students declare more frequently that those not trained, in a nutshell? Do not repeat figures already given in tables, unless they are especially important.

We tried to shorten the text by deleting redundant information. The text does not contain data presented in tables or figures, it only summarizes the main findings and describes data which are not presented either in tables nor as figures.

9. DISCUSSION. The first two paragraphs have, in my opinion, little added value, as the ideas here have been already said in previous sections. I’d rather begin the summary of findings in the third paragraph. Conversely, the discussion lack a section comparing the results obtained with previuosly published studies.

We deleted the first two paragraphs of the Discussion, as suggested by the Reviewer. We added a paragraph comparing this survey with other similar surveys, as suggested.

10. Conclusions: I think it would be important to also underline in your conclusions that: 1) A vast majority of students show a very positive attitude towards EBM; 2) However, only a minority of students considered their EBM skills as good or advanced.

We added to the Conclusion, as advised by the Reviewer: “Although the attitude towards EBM is generally positive, only a small minority of students rated their EBM-related skills as advanced in the present survey.”

---

## [Decision Letter · Decision Letter 1]

11 Nov 2019

Self-reported attitudes, knowledge and skills of using evidence-based medicine in daily health care practice: a national survey among students of medical and health sciences faculties in Hungary

PONE-D-19-21546R1

Dear Dr. Lohner,

We are pleased to inform you that your manuscript has been judged scientifically suitable for publication and will be formally accepted for publication once it complies with all outstanding technical requirements.

With kind regards,

Cesario Bianchi

Academic Editor

PLOS ONE

Additional Editor Comments (optional):

Dear Dr Lohner,

Thank you for carefully revise your interesting and timely manuscript. It is now acceptable for publication.

If you agree, however, ask a native English to further edit the manuscript before publishing. It is just a suggestion.

Reviewers' comments:

Reviewer's Responses to Questions

**Comments to the Author**

1. If the authors have adequately addressed your comments raised in a previous round of review and you feel that this manuscript is now acceptable for publication, you may indicate that here to bypass the “Comments to the Author” section, enter your conflict of interest statement in the “Confidential to Editor” section, and submit your "Accept" recommendation.

Reviewer #1: All comments have been addressed

Reviewer #2: All comments have been addressed

2. Is the manuscript technically sound, and do the data support the conclusions?

Reviewer #1: Yes

Reviewer #2: (No Response)

3. Has the statistical analysis been performed appropriately and rigorously? 

Reviewer #1: Yes

Reviewer #2: (No Response)

4. Have the authors made all data underlying the findings in their manuscript fully available?

Reviewer #1: Yes

Reviewer #2: (No Response)

5. Is the manuscript presented in an intelligible fashion and written in standard English?

Reviewer #1: No

Reviewer #2: (No Response)

6. Review Comments to the Author

Reviewer #1: (No Response)

Reviewer #2: (No Response)

7. PLOS authors have the option to publish the peer review history of their article (what does this mean?). If published, this will include your full peer review and any attached files.

Reviewer #1: No

Reviewer #2: Yes: Carmelo Lafuente-Lafuente

---

## [Editor Report · Acceptance letter]

10 Dec 2019

PONE-D-19-21546R1 

Self-reported attitudes, knowledge and skills of using evidence-based medicine in daily health care practice: a national survey among students of medicine and health sciences in Hungary 

Dear Dr. Lohner:

I am pleased to inform you that your manuscript has been deemed suitable for publication in PLOS ONE. Congratulations! Your manuscript is now with our production department. 

With kind regards,

on behalf of

Dr. Cesario Bianchi 

Academic Editor

PLOS ONE